# Language-Conditioned Imitation Learning with Base Skill Priors under Unstructured Data

## Abstract

The growing interest in language-conditioned robot manipulation aims to develop robots capable of understanding and executing complex tasks, with the objective of enabling robots to interpret language commands and manipulate objects accordingly. While language-conditioned approaches demonstrate impressive capabilities for addressing tasks in familiar environments, they encounter limitations in adapting to unfamiliar environment settings. In this study, we propose a general-purpose, language-conditioned approach that combines base skill priors and imitation learning under unstructured data to enhance the algorithm's generalization in adapting to unfamiliar environments. We assess our model's performance in both simulated and real-world environments using a zero-shot setting. In the simulated environment, the proposed approach surpasses previously reported scores for CALVIN benchmark, especially in the challenging Zero-Shot Multi-Environment setting. The average completed task length, indicating the average number of tasks the agent can continuously complete, improves more than 2.5 times compared to the state-of-the-art method HULC. In addition, we conduct a zero-shot evaluation of our policy in a real-world setting, following training exclusively in simulated environments without additional specific adaptations. In this evaluation, we set up ten tasks and achieved an average 30% improvement in our approach compared to the current state-of-the-art approach, demonstrating a high generalization capability in both simulated environments and the real world. For further details, including access to our code and videos, please refer to our supplementary materials.

## 1 Introduction

Language-conditioned robot manipulation is an emerging field of research at the intersection of robotics, natural language processing, and computer vision. This domain seeks to develop robots capable of understanding their surrounding environments and executing complex manipulation tasks based on natural language commands provided by humans. Substantial progress has been made in recent years, with some studies focusing on deep reinforcement learning techniques to shape reward functions for language instructions, enabling agents to solve tasks through trial-and-error processes by following language instructions Bahdanau et al. (2018) Nair et al. (2022) Goyal et al. (2021) Bing et al. (2023). However, reinforcement learning approaches often confront limitations due to low sample efficiency and the requirement for careful reward development to learn, which poses challenges in obtaining sufficient training data for effective learning. Consequently, other researchers have turned to language-conditioned imitation learning approaches, which train agents using demonstration datasets to overcome the limitations associated with reinforcement learning. For instance, some studies utilize imitation learning with expert demonstrations that are accompanied by labeled language instructions to solve such language-conditioned tasks Stepputtis et al. (2020), Jang et al. (2021). While these methods have demonstrated a high success rate in completing tasks, there still exists two main shortcomings. Firstly, the process is limited by the substantial effort required to sample expert demonstrations. As a result, the dataset available for exploration of various scenarios in the environment is restricted, ultimately hindering the agent's potential for better performance. Secondly, the trained agent is deficient in its capacity for generalization, which impedes its ability to carry out tasks in unseen environments.

To address the first problem, some researchers employ unstructured data (play data) Lynch et al. (2019) Lynch & Sermanet (2021), which consists of human demonstrations driven by curiosity or other intrinsic motivations, rather than being driven by specific tasks, to reduce the effort required to collect expert data for training. A substantial 99% of the training data is comprised of play data, obtained through interactions with simulation environments by participants using virtual reality (VR) equipment. Only 1% of the data is labeled with language instructions. By employing play data, the labor-intensive task of data labeling is significantly reduced, facilitating the creation of larger training datasets for imitation learning. The trained agent demonstrates remarkable performance, exhibiting a high success rate across various tasks. Building upon the ideas presented in Lynch & Sermanet (2021), HULC Mees et al. (2022a) was developed to enhance the performance of language-conditioned imitation learning by integrating a transformer encoding structure Vaswani et al. (2017) and contrastive representation learning.

Regarding to the second problem, current approaches still face a challenge in generalizing to perform tasks in unfamiliar and complex environment. The policy learned through the imitation learning algorithm exhibits outstanding evaluation performance primarily in training domains, suggesting that the policy's effectiveness is restricted to scenarios where training and evaluation environments are identical. Upon conducting sim2real experiments and zero-shot evaluations in novel environments, the discrepancy between the evaluation and training environments result in a substantial decline in success rates. The Zero-shot Multi-environment evaluation provided by the CALVIN benchmark Mees et al. (2022b) further highlights the limitations of current language-conditioned imitation learning approaches (the success rate decreased by 50 per cent), illustrating that trained agents struggle to ground language instructions to target objects and actions in unfamiliar environments.

Within the framework of imitation learning, agents typically rely on predicting the short-term next action at each time step based on the current observation and goal without learning a high-level long-term procedure. This approach diverges the more natural approach employed by humans, which typically involves breaking down complex tasks into simpler, basic steps. Skill-based learning Shi et al. (2022) Nagabandi et al. (2020) is a promising approach that utilizes pre-defined skills to expedite the learning process, leveraging the prior knowledge encoded within these skills, which is typically derived from human expertise. A primary factor contributing to the suboptimal performance of current language-conditioned imitation learning methodologies is the absence of prior knowledge during the training process. The excessive dependence on training data can lead to overfitting and impede generalization to unfamiliar scenarios. By incorporating prior skills into the learning process, the agent can avoid the necessity to start from scratch and reduce the dependency of training data.

In this paper, we introduce a base **S**kill **P**rior based **I**mitation **L**earning (**SPIL**) framework designed to enhance the generalization ability of an agent in adapting to unfamiliar environments by integrating base skill priors: translation, rotation, and grasping. Specifically, SPIL learns both a low-level policy for skill instance execution based on observations, as well as a high-level policy that determines which base skill (translation, rotation, and grasping) should be performed under the current observation. The high-level policy functions as a manager, interpreting language instructions and appropriately combining these base skills to solve complex manipulation tasks. For instance, when the high-level policy receives the language instruction "lift the block", it will decompose the task into several steps involving base skills, such as approaching the block (translation), grasping the block (grasping), and lifting the block (translation). We evaluate our algorithm using the CALVIN benchmark and achieve state-of-the-art performance in both the Single Environment and the challenging Zero-shot Multi Environment settings. Furthermore, we conduct sim-to-real experiments to assess the performance of our approach in real-world environments, yielding outstanding results. We summarize the key contributions as follows:

- In this paper, we incorporate the skill priors into imitation learning and carefully crafted a skill-prior-based imitation learning mechanism to enable learning of a high-level procedure and enhance the generalization ability of the learned policy.

- Our proposed method exhibits superior performance compared to previous baselines, particularly in terms of its ability to generalize and perform well in previously unseen environments. Our evaluation shows that our approach outperforms the current state-of-the-art method by a significant margin, achieving 2.5 times the performance. We conducted a series of sim-to-real experiments to further investigate the generalization ability of our model in unseen environments and the potential of our model for real-world applications.

## 2 RELATED WORKS

Recently, natural language processing has attracted significant interest and attention within the field of robotics, grounding language to behaviors based on the vision observations. This section covers the popular language-robot manipulation framework. Additionally, the skill-based learning methods which inspired our novel approach are also discussed.

In this field, some studies concentrate on establishing connections between visual perception and linguistic comprehension in the vision-and-language field, facilitating the agent's ability to tackle multimodal problems Pont-Tuset et al. (2020) Lu et al. (2019) Li et al. (2020). While other research focuses on grounding language instructions and the agent's behaviors, empowering the agent to comprehend instructions and effectively interact with the environment Shridhar et al. (2020) Magassouba et al. (2019) Liu et al. (2022) Shridhar et al. (2022). However, these approaches employ two-stream architectural models for the processing of multimodal data. Such model require distinct feature representations for each data modality, such as semantic and spatial representations Shridhar et al. (2022), thus potentially compromising learning efficiency. As an alternative, end-to-end models focus on learning feature representations and decision-making directly from raw input data, where the language instructions as a conditioning factor to train the agent. This approach eliminates the need for manual feature engineering Mees et al. (2022a)Co-Reyes et al. (2018), thereby offering a more efficient and robust solution for complex tasks and emerging as a trend in the field of language-conditioned robot manipulations.

For instance, imitation learning with end-to-end models has been applied to solve language-conditioned manipulation tasks using expert demonstrations accompanied by a large number of labeled language instructions Stepputtis et al. (2020) Jang et al. (2021). These approaches necessitate a substantial amount of labeled and structured demonstration data. By extending the idea of Lynch et al. (2019), Lynch et al. proposed MCIL Lynch & Sermanet (2020), which grounds the agent's behavior with language instructions using unlabeled and unstructured demonstration data, reducing data acquisition efforts and achieving more robust performance. HULC Mees et al. (2022a), as an enhanced version of MCIL, designed to improve the performance of MCIL even further. It has achieved impressive results in the CALVIN benchmark Mees et al. (2022b) using the single environment setting. However, when tested in the more challenging Zero-shot Multi Environment setting, where the evaluation environment is not exactly same as the training environments, HULC's performance drops significantly. These suboptimal results suggest that current language-conditioned imitation learning approaches lack the ability to adapt to unfamiliar environments.

The concept of skill-based mechanisms in deep reinforcement learning provides valuable insights for enhancing the generalizability of algorithms. Specifically, skill-based reinforcement learning leverages task-agnostic experiences in the form of large datasets to accelerate the learning process Hausman et al. (2018) Merel et al. (2019) Merel et al. (2019) Kipf et al. (2019) Lee et al. (2020). To extract skills from a large task-agnostic dataset, several approaches Pertsch et al. (2021) Pertsch et al. (2020) first learn an embedding space of skills and skill priors from the dataset. Taking inspiration from this, we have developed an imitation learning approach that utilizes certain base skill priors. By employing this method, the agent learns high-level processes (composing these base skills) that aid in task completion, thereby enhancing its ability to generalize across different scenarios.

## 3 METHODOLOGY

In this section, we first provide an overview of our approach. Following that, we introduce the details of our skill-prior-based imitation learning, which includes the method to construct a continuous skill embedding space with base skill priors and the mechanism to integrate base skill priors into imitation learning.

### 3.1 OVERVIEW

The optimization strategy employed in imitation learning involves minimizing the discrepancy between the predicted actions and the corresponding actions observed in the demonstration data. A primary challenge in integrating skill priors into imitation learning is the continuous nature of actions in the demonstration data, which requires modeling the skills as a continuous action space to

align with the demonstration actions, rather than representing the skills by a finite, discrete set of pre-defined action sequences.

Our goal is to break down complex tasks into simpler steps. We define three base skills for the robot arm agent, namely, translation, rotation, and grasping, as these constitute the most basic behaviors of such an agent. These base skills are integrated into the continuous skill space by introducing three base skill distributions in the skill space.

By utilizing a continuous skill space and base skills, we implement an imitation learning algorithm to train the agent to acquire the ability to 1) learn a high-level base skill composition to accomplish the desired task and 2) develop a policy that can determine the appropriate skill instance to perform based on each observation, as opposed to a single action. In other words, the action space of the agent is transformed into a skill embedding space. The architecture of our proposed method is illustrated in Figure 2.

## 3.2 LEARNING CONTINUOUS SKILL EMBEDDINGS AND INTEGRATING BASE SKILLS

### 3.2.1 BASE SKILL CLASSIFICATION

Considering that the base skills encompass translation, rotation, and grasping, it is feasible to create a straightforward classification model manually to categorize a specific action sequence to its corresponding base skill type. This classification can be accomplished by assessing the accumulated magnitude of seven degrees of freedom within the temporal dimension of a given horizon $H$. For a given action sequence $(a_t, a_{t+1}, ..., a_{t+H-1})$, the probability of this sequence belonging to translation, rotation, and grasping skill can be defined as follows:

$$p(y = k|x) = \frac{w_k \cdot \sum_{i=t}^{t+H-1} |a_i^k|}{\sum_{k \in \{\text{trans., rot., grasp.}\}} w_k \cdot \sum_{i=t}^{t+H-1} |a_i^k|}, \tag{1}$$

where $k \in \{\text{trans.,rot.,grasp.}\}$ and $a^k$ is the action's corresponding change of $k$. $w_k$ is the weight parameter to balance the inconsistency in the scale.

### 3.2.2 CONTINUOUS SKILL EMBEDDINGS WITH BASE SKILL PRIORS

We define $y$ as the indicator for base skills, so that the base skill distribution in the latent space can be written as $z \sim p(z|y) = \mathcal{N}(\mu_y, \sigma_y^2)$. By extending the idea of VAE, we incorporate the variable $y$ into the model. We employ the approximate variational posterior $q(z|x)$ and $q(y, z|x)$ to estimate the intractable true posterior. Following the the VAE procedure, we measure the Kullback-Leibler (KL) divergence between the true posterior and the posterior approximation to determine the ELBO (detailed proof can be found in the appendix A.5.1):

$$\mathcal{L}_{\text{ELBO}} = \overbrace{\mathbb{E}_{z \sim q_\phi(z|x)}[\log p_\theta(x|z)]}^{\text{reconstruction loss}} - \beta_1 \overbrace{D_{KL}(q_\phi(z|x)||p(z))}^{\text{regularizer}} \\ - \beta_2 \sum_k q(y = k|x) \underbrace{D_{KL}(q_\phi(z|x)||p_\kappa(z|y = k))}_{\text{base-skill regularizer}}, \tag{2}$$

where $p_\theta(x|y, z)$ and $q_\phi(z|x)$ are the decoder and encoder networks with parameters $\theta$ and $\phi$, respectively. We also define a network $p_\kappa(z|y)$ with parameters $\kappa$ for locating the base skills in the latent skill space. Note that we introduce hyperparameters $\beta_1$ and $\beta_2$ to weigh the regularizer terms. $\mathcal{L}_{\text{ELBO}}$ can be interpreted as follows. On the one hand, we intend to achieve higher reconstruction accuracy. As the reconstruction improves, our approximated posterior will become more accurate as well. On the other hand, the two introduced regularizers contribute to a more structured latent skill space. The first regularizer, $D_{KL}(q_\phi(z|x)||p(z))$, constrains the encoded distribution to be close to the prior distribution $p(z)$. Likewise, the second regularizer, $D_{KL}(q_\phi(z|x)||p_\kappa(z|y))$, draws the encoded distribution nearer to the prior distribution of its corresponding base skill class.

The learning procedure is illustrated in Figure 1. After the training process, we obtain a skill generator, $\mathcal{G} = p_\theta(x|z)$, which maps the skill embedding to the corresponding action sequence. Additionally, we have the base skill locator $\mathcal{B} = p_\kappa(z|y)$ to identify the position of base skill distributions within the skill latent space. Their parameters are frozen during later imitation learning process.

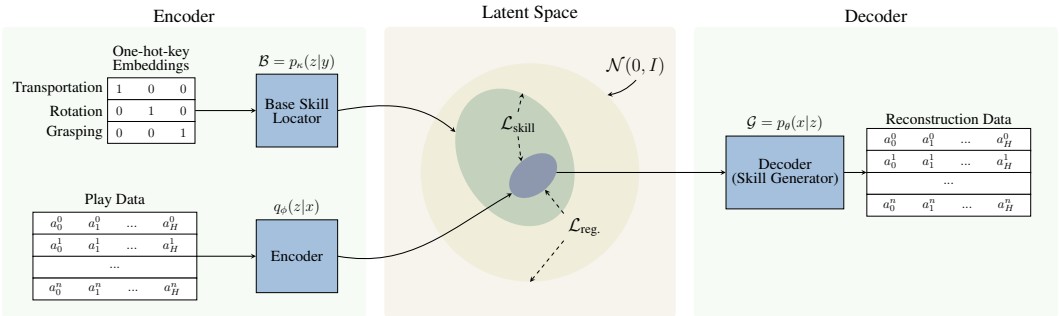

Figure 1: This architecture comprises two encoders - the action sequence encoder and the base skill locator (encoder), and a decoder for reconstructing the skill embeddings into action sequences. The base skill locator takes one-hot-key embeddings of translation, rotation, and grasping as input and outputs the distribution of the base skill prior in the skill latent space. The action sequence encoder encodes the action sequences with a fixed horizon of $H$ to the distribution of skill in the latent space. The decoder then reconstructs the skill embedding into action sequences.

## 3.3 IMITATION LEARNING WITH BASE SKILL PRIORS

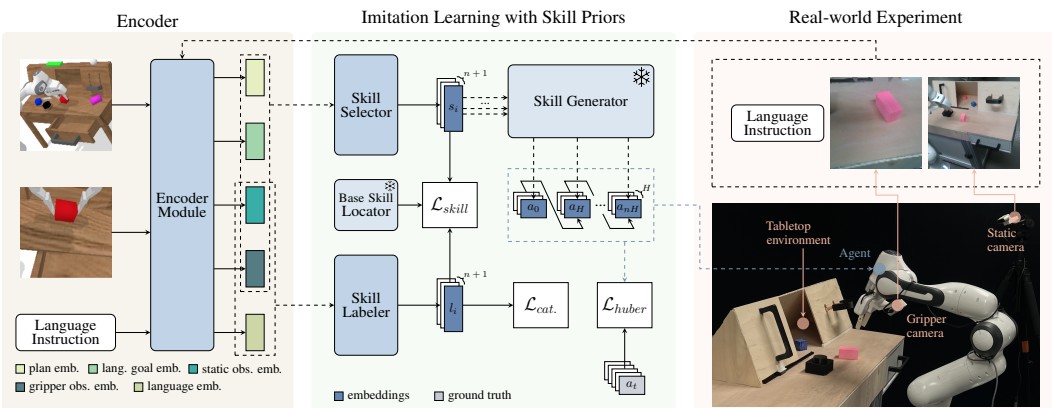

Figure 2: The architecture of the proposed method. Following the encoding process, the static observation, gripper observation, and language instruction are generated to embeddings for the plan, language goal, language, static observation, and gripper observation. The skill selector module subsequently decodes a sequence of skill embeddings using the plan, observation, and language goal embeddings. The skill labeler is responsible for labelling the skill embeddings with the base skills: translation, rotation, and grasping. The base skill regularization loss is calculated based on the base skill prior distributions (from base skill locator $\mathcal{B}$), selected skill instance, and labelled probability indicating its belonging to specific base skills. This labelled probability is also leveraged to determine the categorical regularization loss. Finally, the pre-trained and frozen skill generator $\mathcal{G}$ decodes all the skill embeddings into action sequences, which are then utilized to calculate the reconstruction loss (huber loss).

In this section, we introduce a novel approach that leverages base skill priors to train a policy using the imitation learning algorithm. This approach results in a policy with enhanced generalization capabilities, as the incorporation of prior knowledge prevents model from overfitting. The prior knowledge can bias the model towards solutions that align with our prior understanding, thus preventing the model from learning irrelevant or noisy patterns in the data that may lead to overfitting. A renowned instance is the $L_1$ or $L_2$ regularization in maximum a posteriori estimation (MAP).

We extend the idea of MCIL Lynch & Sermanet (2021) and HULC Mees et al. (2022a) by employing an action space comprising skill embeddings instead of Cartesian action space. In this framework, the action performed by the agent is no longer a single 7 DoF movement in one time step, but instead, a skill (action sequence) over a horizon $H$. Consequently, the agent learns to select a skill based on the current observation. After the skill is performed, the agent selects the next skill based on the subsequent observation, and the process continues iteratively until the agent completes the task or the

---

**Algorithm 1** Imitation Learning with Skill Priors

---

1: Given:
   - $\mathcal{D} : \{(D^{\text{play}}, D^{\text{lang}})\}$: Play Dataset and Language Dataset
   - $\mathcal{F} = \{f_\phi, f_\lambda, f_\kappa, f_\omega, f_\theta\}$. They are the encoder network with parameters $\phi$, the base skill selector network with parameters $\lambda$, the base skill locator network with parameters $\kappa$, the skill labeler network with parameters $\omega$, and the skill generator with parameters $\theta$.
2: Randomly initialize model parameters $\{\phi, \lambda, \omega\}$
3: Initialize parameters $\theta$ and $\kappa$ with pre-trained skill generator and base skill locator and freeze the parameters.
4: **while** not done **do**
5:     $\mathcal{L} \leftarrow 0$
6:     **for** $k$ in {play, lang} **do**
7:         Sample a (demonstration, context) batch from the dataset $D$: $(\tau^k, c^k) \sim \mathcal{D}^k$
8:         Encode the observation, goal, plan embeddings, using the encoder network $f_\phi$
9:         **Skill Selecter** $f_\lambda$ selects the skill embedding sequence
10:        Determinate a sequence of base skills with **Skill Labeller** $f_\omega$.
11:        Determinate base skill locations in the latent space with **Base Skill Locator** $f_\kappa$
12:        **Skill Generator** $f_\theta$ maps the skill embeddings to action sequences.
13:        Calculate the loss function $\mathcal{L}_k$ according to equation 4
14:        Accumulate imitation loss $\mathcal{L} \mathrel{+}= \mathcal{L}_k$
15:     **end for**
16:     update parameters $\{\phi, \lambda, \omega\}$ by taking a gradient step w.r.t $\mathcal{L}$
17: **end while**

---

time runs out. Figure 2 depicts the overall structure of our approach. Given the superior performance of the HULC model, we employ its encoder to transform the static observation, gripper observation, and language instruction into their corresponding embeddings. It is important to note that we extract two embeddings from language instruction. The first is the language goal embedding, which lies in the same latent goal space as image goal embedding. The second is the language embedding, which is utilized to infer the appropriate base skill to be employed, given the current observation. Revisiting the concept of multi-context learning, our objective is to train a single imitation learning policy conditioned on either language instruction or goal observation. Therefore, the encoded language embedding and goal image embedding should be in the same space. However, this structure presents a challenge, as the language instruction contains not only the final state information but also the process information about how to complete the tasks. This process information is crucial for inferring the high-level compositions of base skills required for successfully task completion. We further analyze four key parts in our structure:

- **Skill Selector**: The skill selector generates skill embeddings in the pre-trained latent space based on the encoded plan embedding, language goal embedding, static observation embedding, and gripper observation embedding. A bidirectional LSTM network is employed for this skill selector.

- **Skill Labeler**: The base skill labeler, also a bidirectional LSTM network, determines the base skill to which a given skill belongs. It is used to label the skill embeddings generated from the Skill Selector.

- **Base Skill Locator**: The base skill locator shares the same parameters with base skill locator $\mathcal{B}$ in Figure 1. It has the task of locating the base skill locations in the latent space which used to calculate the regularization loss.

- **Skill Generator**: The skill generator, denoted as $\mathcal{G} = p_\theta(x|z)$, is the decoder component in Figure 1. Its parameters are frozen during the training process. This generator decodes the skill embeddings into action sequences that are combined in chronological order.

The objective of our model is to learn a policy $\pi(\tau_t|s_c, s_g)$ conditioned on the current state $s_c$ and the goal state $s_g$, and outputting $\tau_t$, a sequence of actions, namely a skill. Given the incorporation of the base skill concept into our model, the policy $\pi(\cdot)$ should also identify the optimal base skill $y$ under the current observation. Therefore, we have $\pi(\tau_t, y_t|s_c, s_g)$, where $y_t$ represents the current

base skill chosen by the agent. Inspired from the conditional variational autoencoder (CVAE) Sohn et al. (2015)

$$\log p(x|c) \geq \mathbb{E}_{q(z|x,c)}[\log p(x|z,c)] - D_{KL}(q(z|x,c)||p(z|c)), \tag{3}$$

where $c$ denotes a general condition, we aim to extend the above equation by integrating $y$. It indicates the base skill to accommodate our model. The evidence we would like to maximize then turns to $p(x, y|c)$. We employ the approximate variational posterior $q(y, z|x, c)$ to approximate the intractable true posterior $p(y, z|x, c)$. We intend to find the ELBO by measuring the KL divergence between the true posterior and the posterior approximation (detailed theoretical motivation can be found in A.5.2).

$$
\mathcal{L}_{\text{ELBO}} = \overbrace{\mathbb{E}_{z \sim q_\phi(x,c)} \log p_\theta(x|z,c)}^{\text{reconstruction loss}} \\
- \gamma_1 \sum_k q_\lambda(y = k|c) \underbrace{D_{KL}(q_\phi(z|x,c)||p_\kappa(z|y))}_{\text{base skill regularizer}} - \gamma_2 \underbrace{D_{KL}(q_\lambda(y|c)||p(y))}_{\text{categorical regularizer}}, \tag{4}
$$

where $c$ represents a combination of the current state and the goal state $(s_c, s_g)$. $z$ is skill embedding in the latent skill space. $p_\theta(x|z,c)$ is the skill generator network $\mathcal{G}$ with parameters $\theta$ and it is pretrained and frozen during the training. $q_\lambda(y|c)$ corresponds to the skill labeller with parameter $\lambda$. $q_\phi(z|x,c)$ refers to the encoder network with parameters $\phi$. Furthermore, $p_\kappa(z|y)$ constitutes the base skill prior locater $\mathcal{B}$ with parameter $\kappa$. It is also pre-trained and frozen during the training process. The overall algorithm can be seen in Algorithm 1.

## 4 EXPERIMENTS

In this section, we present the details of the experiment conducted to investigate the generalization ability of our model in comparison to other baselines. We provide a comprehensive description of the experimental environment we used. The results of the experiment are discussed in the following section.

### 4.1 ENVIRONMENT SETTING

Mees et al. introduce CALVIN benchmark Mees et al. (2022b) to facilitate learning long horizon language-conditioned tasks across four manipulation environments. These environments are designed to be diverse while maintaining shared structure. They provide observations from multimodel onboard sensor. In our setting, the agent, a Franka Emika robot arm equipped with a gripper end effector, should complete complex manipulation tasks by understanding a sequence of unconstrained language expressions in a row. This benchmark only allows feasible sequences that can be achieved from a predefined initial environment state. We choose **Long-Horizon Multi-Task Language Control** (LH-MTLC) to evaluate the effectiveness of the learned multi-task language-conditioned policy in accomplishing several language instructions in a row under the **Single Environment** and **Zero-shot Multi Environment**. Comparison with skill-based reinforcement learning approaches can be found in appendix A.4.

### 4.2 RESULTS AND ABLATION STUDIES

Table 1: CALVIN benchmark results (single environment)

| Method | Train → Test | LH-MTLC | | | | | |
|---|---|---|---|---|---|---|---|
| | | No. Instructions in a Row (1000 chains) | | | | | |
| | | 1 | 2 | 3 | 4 | 5 | Avg. Len. |
| LangLfp | D → D | 76.4 % (1.5) | 48.8% (4.1) | 30.1 % (4.5) | 18.1% (3.0) | 9.3 % (3.5) | 1.82 (0.2) |
| HULC | D → D | 82.7% (0.3) | 64.9% (1.7) | 50.4 % (1.5) | **38.5** % (1.9) | 28.3 % (1.8) | 2.64 (0.05) |
| SPIL (Ours) | D → D | **84.6**% (0.6) | 65.1% (1.3) | **50.8** % (0.4) | 38.0 % (0.6) | **28.6** % (0.3) | **2.67** (0.01) |
| $\gamma_1 = 1.0 \times 10^{-2}$ | D → D | 83.9% (0.4) | **65.4**% (0.5) | 49.1 % (1.0) | 35.4 % (1.1) | 26.4 % (0.8) | 2.60 (0.3) |
| $\gamma_2 = 1.0 \times 10^{-4}$ | D → D | 84.5% (1.0) | 64.8% (2.0) | 47.5 % (2.2) | 34.5 % (1.5) | 24.0 % (0.4) | 2.55 (0.06) |

We analyze the result of our model by comparing it to other baselines (shown in Table 1, 2). We evaluate the models with 1000 five-tasks chains. The columns labeled from one to five demonstrate the success rate to continuously complete that number of tasks in a row. The average length indicates

Table 2: CALVIN benchmark results (zero-shot multi environment)

| Method | Train → Test | LH-MTLC | | | | | |
|---|---|---|---|---|---|---|---|
| | | No. Instructions in a Row (1000 chains) | | | | | |
| | | 1 | 2 | 3 | 4 | 5 | Avg. Len. |
| LangLfP | A,B,C → D | 30.4% | 1.3% | 0.17 % | 0 % | 0 % | 0.31 |
| HULC | A,B,C → D | 41.8% (2.3) | 16.5% (2.5) | 5.7% (1.3) | 1.9 % (0.9) | 1.1 % (0.5) | 0.67 (0.1) |
| SPIL (Ours) | A,B,C → D | **74.2%** (1.4) | **46.3%** (3.4) | **27.6** % (3.4) | **14.7** % (2.3) | **8.0** % (1.7) | **1.71** (0.11) |
| $\gamma_1 = 1.0 \times 10^{-2}$ | A,B,C → D | 71.3% (1.4) | 45.8% (3.8) | 25.4 % (2.3) | 13.1 % (0.9) | 6.5 % (0.5) | 1.62 (0.05) |
| $\gamma_2 = 1.0 \times 10^{-4}$ | A,B,C → D | 70.6% (4.2) | 46.3% (3.2) | 25.1 % (3.0) | 14.1 % (1.0) | 7.3 % (1.3) | 1.63 (0.08) |

the average number of tasks the agent can continuously complete when given five tasks in a row (The remaining tasks are not performed if one task fails in the middle). Subsequently, ablation studies on hyperparameters $\gamma_1,\gamma_2$ in equation 4 are performed both in Single Environment and Zero-Shot Multi Environment. Our model achieves state-of-the-art (SOTA) results, outperforming all baselines in the CALVIN benchmark, especially in the challenging Zero-Shot Multi Environment. Each model is evaluated three times across 3 random seeds.

### 4.2.1 SINGLE ENVIRONMENT

As shown in Table 1, our model achieves the highest score among all baselines in the Single Environment setting. Compared to the current SOTA model HULC, the success rate of completing one to five tasks in a row has increased by 1.9 %, 0.2%, 0.4 %, -0.5%, and 0.3 % respectively. The overall average completed task length, indicating the average number of continuously completed tasks, increased from 2.64 to 2.67. It is important to note that this experiment demonstrates that in the single environment setting, our proposed method does not exhibit any performance degradation (even higher) compared to other baselines. The primary focus of our work is on the Zero-shot Multi Environment setting, which serves to demonstrate the model's generalization ability.

### 4.2.2 ZERO-SHOT MULTI ENVIRONMENT

As evidenced in Table 2, our model achieves substantial improvement compared to our baselines HULC and LangLfP. In comparison to the current SOTA model HULC, the success rate of completing one to five tasks in a row has increased by 32.4%, 29.8%, 21.9 %, 12.8%, and 6.9 %, respectively. The overall average length increased from 0.67 to 1.71. It is worth noting that the zero-shot Multi environment presents a considerable challenging environment, as the agent is required to solve tasks under an unfamiliar environment. The performance in this setting represents the agent's generalization ability, in other words, the ability to truly understand and connect the concepts in language instructions with real objects and actions. The performance of our model demonstrates a significant improvement in generalization ability, thus confirming our hypothesis that using skill priors to learn high-level task composition can improve generalization capabilities.

### 4.3 REAL-WORLD EXPERIMENTS

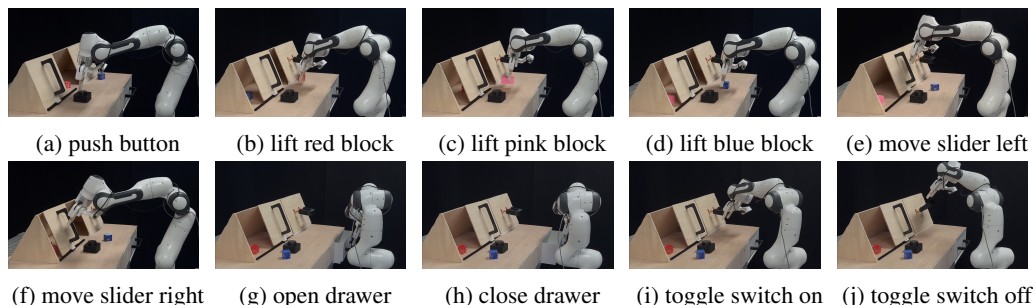

   (a) push button    (b) lift red block    (c) lift pink block    (d) lift blue block    (e) move slider left

   (f) move slider right    (g) open drawer    (h) close drawer    (i) toggle switch on  (j) toggle switch off

Figure 3: Real-world experiments.

To investigate the viability of the policy trained in a simulated environment to real-world scenarios, we conduct a sim2real experiment without any additional specific adaptation, as shown in Figure 3.

We designed the real-world environment to closely resemble the simulated environment. The rightmost part of Figure 2 illustrates that the real-world environment comprises one switch, one cabinet with a slider, one button, one drawer, and three blocks in red, pink, and blue colors. Additionally, two RGB cameras are employed to capture the static observation and gripper observations.

Table 3: Real-world experiment results

| Tasks | HULC | SPIL |
|---|---|---|
| open drawer | 0% | 30% |
| close drawer | 0% | 40% |
| toggle switch on | 10% | 40% |
| toggle switch off | 10% | 30% |
| move slider left | 0% | 40% |
| move slider right | 0% | 30% |
| push button | 10% | 50% |
| lift red block | 0% | 20% |
| lift blue block | 0% | 20% |
| lift pink block | 0% | 30% |
| Average | 3% | 33% |

The tasks performed and the corresponding success rate are listed in Table 3. The agent is trained with four CALVIN environments (A,B,C,D) and the trained policy is directly applied to a real world environment. To mitigate the influence of the robot's initial position on the policies, we execute 10 roll-outs for each task, maintaining identical starting positions. The results in the table demonstrate the effectiveness of our model in handling the challenging zero-shot sim2real experiments. Despite the substantial differences between the simulation and real-world contexts, our model still achieves an average success rate of 33% in accomplishing the tasks. Conversely, the HULC model-trained agent struggles with these tasks, with a 3% average success rate, underscoring the difficulty of solving real-world challenges. The results from real-world experiments further substantiate our claim that our proposed method exhibits superior generalization capabilities, enabling successful task completion even in unfamiliar environments.

## 5 DISCUSSION

In this paper, we propose a novel imitation learning paradigm that incorporates base skill priors to enhance the generalization ability of the trained agent in unseen environment. The fundamental concept of our approach is to acquire a high-level base skill decomposition of tasks that aligns with the natural way in which humans typically perform tasks. The variational autoencoder (VAE) procedure employed for processing the action sequences also serves to reduce their dimensionality, thereby mitigating the well-known curse of dimensionality to a certain extent. This dimensionality reduction leads to an improved predictive capacity for the neural network. It is also noteworthy that our proposed approach is a generic method (a variant of imitation learning) that may also be applied in other robot arm manipulation tasks to increase the generalization ability.

Despite that our approach achieves much better generalization ability compared to previous baselines, there still exists some limitations warrant further research. For instance, we found that the utilization of a gripper camera and static camera setting presents a challenge for the agent to accurately perceive the distance between the end-effector and the target object. The disparities between the real-world experimental environment and the simulated experimental environment, such as variances in object size, shape, and camera angles and positions, can create difficulties for the agent when attempting to infer the actual location relationship between its gripper and the objects. We speculate that incorporating a binocular camera or depth camera near the gripper may aid the agent in extracting more precise distance information.

## 6 CONCLUSION

In this paper, we introduced a novel imitation learning paradigm that integrating base skills into the imitation learning. Our proposed SPIL model effectively improves the generalization ability compared to current baselines and substantially surpasses the state-of-the-art models on the language-conditioned robotic manipulation CALVIN benchmark, especially under the challenging Zero-Shot Multi Environment setting. This work also aims to contribute towards the development of general-purpose robots that can effectively integrate human language with their perception and actions.

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

## A APPENDIX

### A.1 CALVIN ENVIRONMENT DETAIL

The CALVIN benchmark affords multiple types of observations, including RGB static camera, depth static camera, RGB gripper camera, depth gripper camera, tactile image, and preoperative state. In our approach, we limit the observation space to RGB static camera and Depth gripper camera. CALVIN's action space consists of absolute cartesian pose, relative cartesian displacement, and joint action. We specifically select relative cartesian displacement from the action space to use in our approach. The environment can be characterized by following statement.

- A multidimensional action space $\mathcal{A} \subset \mathbb{R}^7$. This action space contains all parameters to drive the agent to finish tasks. The first three parameters are the displacement of the end effector's position. Another three parameters are the rotation of the end effector and the final parameter is the gripper control parameter.

- A visual state space $\mathcal{S} \subset \mathbb{R}^{N \times H \times W \times 3}$. $N$ is the number of observations. $H$ and $W$ are the height and width of the images since the agent only has access to the visual observations from cameras.

- A multi-context goal space consists of language instructions or goal images $\mathcal{G} \subset \mathcal{L} \cup \mathbb{R}^{H \times W \times 3}$, where $\mathcal{L}$ is the natural language set and $H$, $W$ are the height and width of the images respectively.

CALVIN benchmark mainly contains three components.

- **CALVIN environments**. CALVIN includes four distinct environments (A, B, C, D) that are interconnected in terms of their underlying structure. Each environment consists of one Franka Emika Panda robot arm equipped with a gripper and a desk featuring a sliding door and a drawer that can be opened and closed. On the desk, there exists a button that can toggle the green light and a switch to control a light bulb. Note that each environment has a different desk with various of textures and the position of static elements such as the sliding door, drawer, light, switch, and button are different across each environment.

- **CALVIN dataset**. To comprehensively explore the possible scenarios within a given space, the individuals involved engaged in teleoperated play while wearing an HTC Vive VR headset for a total of 24 hours, spending roughly the same amount of time (6 hours) in each of four different environments. In terms of language instructions, they utilize 400 natural language instructions that correspond to over 34 different tasks to label episodes in a procedural manner, based on the recorded state of the environment in the CALVIN dataset.

- **CALVIN challenge**. The authors of the CALVIN introduce various evaluation protocols and metrics of different difficulty levels. These protocols are

  - **Single Environment:** Training in a single environment and evaluating the policy in the same environment.
  - **Multi Environment:** Training in all four environments and evaluating the policy in one of them.
  - **Zero-Shot Multi Environment:** This involves training the agent in three different environments and then testing its ability to generalize and perform well in a fourth environment that it has not previously encountered.

  and the metrics are

  - **Multi-Task Language Control (MTLC):** The most straightforward evaluation aims to verify how well the learned multi-task language-conditioned policy generalizes 34 manipulation tasks
  - **Long-Horizon Multi-Task Language Control (LH-MTLC):** In this evaluation, the 34 tasks from a previous evaluation are treated as subgoals, and valid sequences consisting of five sequential tasks are computed.

Table 4: Important hyperparameters (Single Environment)

| Description | Value |
|---|---|
| Batch Size | 64 |
| Learning Rate | $1.0 \times 10^{-4}$ |
| Skill Embedding Dimension | 20 |
| Horizon Length $H$ of Skill | 5 |
| Magic Scales $w_1, w_2, w_3$ | 1.4, 3.0, 0.75 |
| Plan prior matching weight $\beta$ | $5.0 \times 10^{-4}$ |
| Regularizer weight $\beta_1$ | $1.0 \times 10^{-4}$ |
| Regularizer weight $\beta_2$ | $1.0 \times 10^{-5}$ |
| Regularizer weight $\gamma_1$ | $5.0 \times 10^{-3}$ |
| Regularizer weight $\gamma_2$ | $1.0 \times 10^{-5}$ |

Table 5: Important hyperparameters (Zero-shot Multi Environment)

| Description | Value |
|---|---|
| Batch Size | 32 |
| Learning Rate | $1.0 \times 10^{-4}$ |
| Skill Embedding Dimension | 20 |
| Horizon Length $H$ of Skill | 5 |
| Magic Scales $w_1, w_2, w_3$ | 1.4, 3.0, 0.75 |
| Plan prior matching weight $\beta$ | $1.0 \times 10^{-4}$ |
| Regularizer weight $\beta_1$ | $1.0 \times 10^{-4}$ |
| Regularizer weight $\beta_2$ | $1.0 \times 10^{-5}$ |
| Regularizer weight $\gamma_1$ | $5.0 \times 10^{-3}$ |
| Regularizer weight $\gamma_2$ | $1.0 \times 10^{-5}$ |

## A.2 IMPLEMENTATION DETAILS

### A.2.1 SKILL EMBEDDING SPACE GENERATION

For Single Environment, the skill embedding space is generated by the action sequences in training data of environment D. Regarding to Zero-shot Multi Environment, the skill embedding space is generated by the action sequences in training data of environment A, B, and C. Important hyperparameters are listed in Table 4 and 5.

### A.2.2 TRAINING SETTING

The hyperparameters leveraged to train the agent in Single Environment and Zero-shot Multi Environment setting are listed in Table 4 and 5. The camera observations are applied with image augmentation strategy. For simulation evaluations, the static observation goes through random shift of 10 pixels and normalization with mean=[0.48145466, 0.4578275, 0.40821073] and std=[0.26862954, 0.26130258, 0.27577711]; the gripper observation goes though random shift of 4 pixels and normalization with the same mean and std as static observation;

For real-world experiments, we apply stronger augmentation for images. The static observation goes through the following transforms, random shift of 10 pixels, color jitter with 0.2 brightness and 0.2 contrast, random rotation of with the range of (-5, 5) degrees, random perspective with distortion-scale 0.1, and finally normalization with mean=[0.48145466, 0.4578275, 0.40821073] and std=[0.26862954, 0.26130258, 0.27577711]. The gripper observation goes through center cropping, random shift with 4 pixels, color jitter with 0.2 brightness and 0.2 contrast, and finally the same normalization as the static observation.

We have also implemented an augmentation strategy for the action sequences, where we randomly set the last three relative actions of a sequence to zero, indicating still actions.

Table 6: Training time in hours

| Environment | LangLfP | HULC | SPIL(ours) |
|---|---|---|---|
| D $\rightarrow$ D | 30 | 42 | 43 |
| ABC $\rightarrow$ D | 102 | 122 | 125 |

## A.3 COMPUTATION TIME

**Hardware and Software**: All of the experiments were performed on a virtual machine with 40 virtual processing units, 356 GB RAM and two Tesla V100 (16GB) GPUs. The virtual machine is equipped with the Ubuntu-20.04-LTS-focal operating system.

Table 6 shows the training time for each model, which was done with 40 epochs for Single Environment (D $\rightarrow$ D) and 30 epochs for Zero-shot Multi Environment (ABC $\rightarrow$ D).

## A.4 COMPARISON WITH OTHER SKILL-BASED APPROACHES

In this section, we intend to compare our model with other popular skill-based learning approaches. The baselines we choose are two skill-based reinforcement learning approaches SpiRL Shi et al. (2022) and SkiMo Pertsch et al. (2020). We assess these methods using a subset of tasks in the CALVIN benchmark, which has been modified to consist of a fixed task chain comprising four assignments, namely Open Drawer - Turn on Lightbulb - Move Slider Left - Turn on LED. This task sequence is evaluated 1000 times to determine the average success rate. The outcomes are presented in the Table 7. Our SPIL approach consistently attains an almost perfect success rate of nearly 100% in this task sequence, outperforming the other baseline methods that utilize skill-based reinforcement learning for agent training. All experiments are evaluated with 3 random seeds.

Table 7: Skill-based approaches

| Model | SpiRL | SkiMO | SPIL(ours) |
|---|---|---|---|
| Avg. Len. (#/4.00) | $3.02 \pm 0.53$ | $3.64 \pm 0.21$ | $\mathbf{3.99} \pm 0.01$ |

## A.5 THEORETICAL MOTIVATION

### A.5.1 CONTINUOUS SKILL EMBEDDINGS WITH BASE SKILL PRIORS

We define $y$ as the indicator for base skills so that the base skill distribution in the latent space can be written as $z \sim p(z|y) = \mathcal{N}(\mu_y, \sigma_y^2)$. By extending the idea of VAE, we integrate the variable $y$ in the model. We employ the approximate variational posterior $q(z|x)$ and $q(y, z|x)$ to approximate the intractable true posterior. Similar to the procedure of VAE, we measure the KL divergence between the true posterior and the posterior approximation to find the ELBO:

$$
\begin{aligned}
D_{KL}(q(y,z|x)||p(y,z|x)) &= \int_y \int_z q(y,z|x) \log \frac{q(y,z|x)}{p(y,z|x)} dzdy \\
&= -\int_y \int_z q(y,z|x) \log \frac{p(y,z|x)}{q(y,z|x)} dzdy \\
&= -\int_y \int_z q(y,z|x) \log \frac{p(x,y,z)}{q(z,y|x)p(x)} dzdy + \log p(x)
\end{aligned}
\tag{5}
$$

$$
D_{KL}(q(z|x)||p(z|x)) = -\int_z q_\phi(z|x) \log \frac{p(z,x)}{q_\phi(z|x)} dz + \log p(x)
\tag{6}
$$

By combining Equation 5 and Equation 6, we have:

$$\log p(x) = \frac{1}{2}\left( \overbrace{\int_y \int_z q(y,z|x)\log\frac{p(x,y,z)}{q(z,y|x)p(x)}dzdy}^{\mathcal{L}_1} + \overbrace{\int_z q_\phi(z|x)\log\frac{p(z,x)}{q_\phi(z|x)}dz}^{\mathcal{L}_2} \right.$$
$$\left. + D_{KL}(q(y,z|x)||p(y,z|x)) + D_{KL}(q(z|x)||p(z|x)) \right) \tag{7}$$

We focus on the ELBO term $\mathcal{L}_{\text{ELBO}} = \frac{1}{2}(\mathcal{L}_1 + \mathcal{L}_2)$:

$$\mathcal{L}_1 = \int_y \int_z q(y,z|x)\log\frac{p(x,y,z)}{q(z,y|x)p(x)}dzdy$$
$$= \int_y \int_z q_\phi(z|x,y)q(y|x)\log\frac{p(x|y,z)p(z|y)p(y)}{q(z,y|x)p(x)}dzdy$$
$$= \int_y q(y|x)\left( \int_z q_\phi(z|x,y)\log p_\theta(x|y,z)dz \right.$$
$$+ \int_z q_\phi(z|x,y)\log\frac{p_\kappa(z|y)}{q_\phi(z|x,y)}dz + \int_z q_\phi(z|x,y)\log\frac{p(y)}{q(y|x)}dz \bigg)dy$$
$$= \int_y q(y|x)\left( \int_z q_\phi(z|x,y)\log p_\theta(x|y,z)dz \right. \tag{8}$$
$$+ \int_z q_\phi(z|x,y)\log\frac{p_\kappa(z|y)}{q_\phi(z|x,y)}dz + \log\frac{p(y)}{q(y|x)} \bigg)dy$$
$$= \int_y q(y|x)\left( \int_z q_\phi(z|x,y)\log p_\theta(x|y,z)dz \right.$$
$$+ \int_z q_\phi(z|x,y)\log\frac{p_\kappa(z|y)}{q_\phi(z|x,y)}dz \bigg)dy + \int_y q(y|x)\log\frac{p(y)}{q(y|x)}dy$$
$$= \int_y q(y|x)\left( \int_z q_\phi(z|x,y)\log p_\theta(x|y,z)dz - D_{KL}(q_\phi(z|x,y)||p_\kappa(z|y)) \right)dy$$
$$- D_{KL}(q(y|x)||p(y))$$

where $p_\theta(x|y,z)$ and $q_\phi(z|y,x)$ are the decoder and decoder network with parameters $\theta$ and $\phi$, respectively. We also define a network $p_\kappa(z|y)$ with parameters $\kappa$ for locating the base skills in the latent skill space. In our setups, the variables $x$ and $z$ are conditionally independent given $y$; the variables $z$ and $y$ are also conditionally independent given $x$ since the encoder and decoder networks do not take $y$ as input. Hence, the above equation can be simplified as:

$$\mathcal{L}_1 = \int_y q(y|x)\left( \int_z q_\phi(z|x)\log p_\theta(x|z)dz - D_{KL}(p_\kappa(z|y)||q_\phi(z|x)) \right)dy - D_{KL}(q(y|x)||p(y))$$
$$= \int_z q_\phi(z|x)\log p_\theta(x|z)dz - \int_y q(y|x)D_{KL}(q_\phi(z|x)||p_\kappa(z|y))dy - D_{KL}(q(y|x)||p(y))$$
$$= \mathbb{E}_{z\sim q_\phi(z|x)}[\log p_\theta(x|z)] - \int_y q(y|x)D_{KL}(q_\phi(z|x)||p_\kappa(z|y))dy - D_{KL}(q(y|x)||p(y))$$
$$\tag{9}$$

We know the variable $y$ is not continuous and has only three possibilities so it can be computed exactly by marginalizing over these three categorical options.

$$\mathcal{L}_1 = \mathbb{E}_{z\sim q_\phi(z|x)}[\log p_\theta(x|z)] - \sum_k q(y=k|x)D_{KL}(q_\phi(z|x)||p_\kappa(z|y=k)) - D_{KL}(q(y|x)||p(y))$$
$$\tag{10}$$

In terms of $\mathcal{L}_2$, we have

$$\mathcal{L}_2 = \mathbb{E}_{z\sim q_\phi(z|x)}[\log p_\theta(x|z)] - D_{KL}(q_\phi(z|x)||p(z)) \tag{11}$$

Then, the total $\mathcal{L}_{\text{ELBO}}$ is formalized as:

$$\mathcal{L}_{\text{ELBO}} = \overbrace{\mathbb{E}_{z \sim q_\phi(z|x)}[\log p_\theta(x|z)]}^{\text{reconstruction loss}} - \beta_1 \overbrace{D_{KL}(q_\phi(z|x)||p(z))}^{\text{regularizer}}$$
$$- \beta_2 \sum_k q(y = k|x) \underbrace{D_{KL}(q_\phi(z|x)||p_\kappa(z|y = k))}_{\text{base-skill regularizer}} \tag{12}$$

Note that we introduce hyperparameters $\beta_1$ and $\beta_2$ to weigh the regularizer terms. Intuitively, Equation 12 is easy to understand. On the one hand, we intend to have a higher reconstruction accuracy. As the reconstruction gets better, our approximated posterior will become more accurate as well. On the other hand, those two introduced regularizers can make the latent skill space more structured. The first regularizer $D_{KL}(q_\phi(z|x)||p(z))$ constrained the encoded distribution close to the prior distribution $p(z)$. The second regularizer $D_{KL}(q_\phi(z|x)||p_\kappa(z|y))$ also pulls the encoded distribution near the prior distribution of its corresponding base skill class.

In our setting, we choose the standard normal distribution $\mathcal{N}(0, I)$ for the prior distribution $p(z)$ and the normal distribution $\mathcal{N}(\mu_\kappa^k, \sigma_\kappa^k)$ for the prior base skill distribution. $\mathcal{L}_2$ is used as the metric for measuring the reconstruction accuracy. According to the supplement above, the Loss function can be formalized as:

$$\mathcal{L} = ||x - \hat{x}||_2 + \beta_1 D_{kl}(q_\phi(z|x)||\mathcal{N}(0, I)) + \beta_2 \sum_k q(y = k|x) D_{KL}(q_\phi(z|x)||p_\kappa(z|y = k)) \tag{13}$$

The algorithm for the whole process is demonstrated in Algorithm 2.

---

**Algorithm 2** Learning Continuous Skill Embeddings with Base Skill Priors

---

1: Given:
- $\mathcal{D} : \{(a_1, a_2, ..., a_H)\}$: A Play dataset full of action sequences with horizon $H$.
- $\mathcal{F} = \{f_\phi, f_\theta, f_\kappa\}$. They are the encoder network with parameters $\phi$, the decoder network with parameters $\theta$, and the base skill locator network with parameters $\kappa$.

2: Randomly initialize model parameters $\{\theta, \phi, \kappa\}$
3: **while** not done **do**
4:     Sample an action sequence $x \sim \mathcal{D}$
5:     Compute the encoded distribution, following the encoder $q_\phi(z|x)$
6:     Compute the base skill distributions, according to $p_\kappa(z|y)$.
7:     Sample one latent embedding $z \sim q_\phi(z|x)$
8:     Feed the sampled $z$ into the decoder $p_\theta(x|z)$ to get the reconstructed action sequence $\hat{x}$
9:     Compute the loss defined in Equation 13
10:     Update parameters $\theta, \phi, \kappa$ by taking the gradient step to minimize $\mathcal{L}$
11: **end while**

---

### A.5.2 IMITATION LEARNING WITH BASE SKILL PRIORS

The objective of our model is to learn a policy $\pi(\tau_t|s_c, s_g)$ conditioned on the current state $s_c$ and the goal state $s_g$ and outputting $\tau_t$, a sequence of actions, namely a skill. Since we introduced the base skill concept into our model, the policy $\pi(\cdot)$ should also find the best base skill $y$ for the current observation. We have $\pi(\tau_t, y_t|s_c, s_g)$, where $y_t$ is the current base skill the agent chooses.
Inspired from the conditional variational autoencoder (CVAE):

$$\log p(x|c) \geq \mathbb{E}_{q(z|x,c)}[\log p(x|z, c)] - D_{KL}(q(z|x, c)||p(z|c)) \tag{14}$$

where c is a symbol to describe a general condition, we would like to extend the above equation by integrating $y$ which indicates the base skill. The evidence we would like to maximize then turns to $p(x, y|c)$. We employ the approximate variational posterior $q(y, z|x, c)$ to approximate the intractable true posterior $p(y, z|x, c)$ where $z$ indicates the skill embeddings in the skill latent space. We intend to find the ELBO by measuring the KL divergence between the true posterior and the posterior

approximation.

$$
\begin{aligned}
D_{KL}(q(y,z|x,c)||p(y,z|x,c)) &= \int_y \int_z q(y,z|x,c) \log \frac{q(y,z|x,c)}{p(y,z|x,c)} dzdy \\
&= -\int_y \int_z q(y,z|x,c) \log \frac{p(y,z|x,c)}{q(y,z|x,c)} dzdy \\
&= -\int_y \int_z q(y,z|x,c) \log \frac{p(x,y,z|c)}{q(z,y|x,c)p(x|c)} dzdy + \log p(x|c)
\end{aligned}
$$
(15)

We focus on the first ELBO term:

$$
\begin{aligned}
\mathcal{L} &= \int_y \int_z q(y,z|x,c) \log \frac{p(x,y,z|c)}{q(z,y|x,c)p(x|c)} dzdy \\
&= \int_y \int_z q(z|x,y,c)q(y|x,c) \log \frac{p(x|y,z,c)p(z|y,c)p(y|c)}{q(z,y|x,c)p(x|c)} dzdy \\
&= \int_y q(y|x,c) \left( \int_z q(z|x,y,c) \log p(x|y,z,c) dz \right. \\
&\quad + \int_z q(z|x,y,c) \log \frac{p(z|y,c)}{q(z|x,c)} dy + \log \frac{p(y|c)}{q(y|x,c)} \right) \\
&= \int_y q(y|x,c) \left( \int_z q(z|x,y,c) \log p(x|y,z,c) dz - D_{KL}(q(z|x,y,c)||p(z|y,c)) \right) dy \\
&\quad - D_{KL}(q(y|x,c)||p(y))
\end{aligned}
$$
(16)

We will examine each element in the equation mentioned and establish a relationship with the model we intend to train.

- variable $x, y, z, c$:
    - $x$: action sequence (skill) the agent chooses, namely $\tau_t$
    - $y$: base skill priors
    - $z$: skill embeddings in the latent skill space
    - $c$: a combination of the current state and the goal state $(s_c, s_g)$
- $q(y|x,c)$: It corresponds to the skill labeler part in our SPIL model. We define this network with parameters $\lambda$ and simplify it as $q_\lambda(y|c)$ by pointing out $x$.
- $q(z|x,y,c)$: It refers to the encoder network with parameters $\phi$, taking $c$ as input in our settings. It can be written as $q_\phi(z|c)$, pointing out $x, y$.
- $p(x|y,z,c)$: It is the decoder network $\mathcal{G}$ with parameters $\theta$. This network only takes $z, c$ as input in our setting and we consider $x$ and $y$ to be conditionally independent given $z, c$. It can then be formalized as $p(x|z,c)$. Note that the parameters of this network are pretrained and frozen during the training process.
- $p(z|y,c)$: It is the base skill prior locater $\mathcal{B}$ with parameter $\kappa$. We assume $z$ and $c$ are conditionally independent given $y$ so that we have $p_\kappa(z|y)$. Note that the parameters of this network are frozen during the training.
- $p(y)$: It is the prior distribution for base skills $y$ which is drawn from a categorical distribution.

Based on the above analysis, the whole equation can be simplified as follows:

$$
\begin{aligned}
\mathcal{L} &= \int_y q_\lambda(y|c) \left( \int_z q_\phi(z|c) \log p_\theta(x|z,c) dz - D_{KL}(q_\phi(z|c)||p_\kappa(z|y)) \right) dy - D_{KL}(q(y|c)||p(y)) \\
&= \int_z q_\phi(z|c) \log p_\theta(x|z,c) dz - \int_y q_\lambda(y|c) D_{KL}(q_\phi(z|x,c)||p_\kappa(z|y)) dy - D_{KL}(q_\lambda(y|c)||p(y)) \\
&= \mathbb{E}_{z \sim q_\phi(z|c)} \log p_\theta(x|z,c) - \sum_k q_\lambda(y=k|c) D_{KL}(q_\phi(z|c)||p_\kappa(z|y)) - D_{KL}(q_\lambda(y|c)||p(y))
\end{aligned}
$$
(17)

By introducing two weights $\gamma_1$ and $\gamma_2$ for the regularization terms, we have

$$\mathcal{L} = \overbrace{\mathbb{E}_{z \sim q_\phi(x,c)} \log p_\theta(x|z,c)}^{\text{Reconstruction loss}}$$
$$- \gamma_1 \sum_k q_\lambda(y = k|c) \underbrace{D_{KL}(q_\phi(z|x,c)||p_\kappa(z|y))}_{\text{Base skill regularizer}} - \gamma_2 \underbrace{D_{KL}(q_\lambda(y|c)||p(y))}_{\text{Categorical regularizer}} \tag{18}$$

We use $\mathcal{L}_{huber}$ loss as the metric for reconstructive loss. The total loss function can be formalized as follows:

$$\mathcal{L} = \mathcal{L}_{huber}(x, \hat{x}) + \gamma_1 \sum_k q_\lambda(y = k|c) D_{KL}(q_\phi(z|x,c)||p_\kappa(z|y))$$
$$+ \gamma_2 D_{KL}(q_\lambda(y|c)||p(y)) \tag{19}$$

Intuitively, the base skill regularizer is used to regularize a skill embedding, depending on its base skill categorial. The categorial regularizer aims to regularize the base skill classification based on the prior categorical distribution of $y$. The overall algorithm can be seen in Algorithm 1.

