# OpenReview forum: "Language-Conditioned Imitation Learning With Base Skill Priors Under Unstructured Data"
_ICLR.cc/2024/Conference — ICLR 2024 Conference Withdrawn Submission_

### Official Review · Reviewer_yTpt · 2023-10-29

**Soundness:** 3 good
**Presentation:** 2 fair
**Contribution:** 2 fair
**Rating:** 3
**Confidence:** 4

**Summary:**

This paper introduces SPIL (base skill prior based imitation learning), which improves generalization of robotic manipulation tasks to new environments by biasing skill embeddings toward one of three predefined base skills for robot manipulation: translation, rotation, and grasping. The encoder learns a continuous skill embedding based on a discrete skill prediction of which of the three discrete skills should be executed, the current image observation (from two angles), and language instruction. A VAE, trained to maximize ELBO, maps H-length action sequences into a continuous skill embedding space and decodes them back into action sequences.

Authors evaluate their method on CALVIN and demonstrate similar to HULC performance on in-domain tasks but stronger performance on out-of-domain tasks. They also evaluate their method zero-shot in the real world where results again outperform HULC.

**Strengths:**

(1) The results look promising on generalization to new envs, outperforming HULC.

(2) Authors set up a real world environment and got good results there over HULC.

(3) Appendix has a derivation for the loss, though I did not look into this carefully.

**Weaknesses:**

Method Weaknesses

(M1) Requires training a classifier on a sum of action magnitudes per dimension over the H-length horizon, which presumably requires ground truth labels on what skill an input action sequence corresponds to. Needing labels (or even needing to tune such a classifier if it were trained in an unsupervised fashion) is a significant limitation and not possible with most robotics datasets.

(M2) Unable to scale to more skills: If a new skill is needed in a new domain, everything would need to be trained from scratch.

(M3) Hard-to-classify action sequences: With explicit skill labeler supervision, it seems hard for SPIL to provide a skill embedding for H-length subtrajectories that have a mix of multiple base skills (such as one that encompasses the transition between translation and grasping an object).

(M4) Skills are blind to the state, the meaningfulness of action sequences (ie: the skill embedding space has no understanding of which skills are good vs just encoding random behavior), and are only as informative as what is expressed in the action magnitudes of each dim. A skill not conditioned on the current state is hard to efficiently adapt to a new domain with a sufficiently different state distribution, as the learning algorithm must determine which skills are appropriate for the current state. For instance, grasping-related skills are not good to execute when an object is already in the robot’s gripper.

(M5) Skill priors have been studied a lot. Behavior Priors (Parrot [1], SKiP [2]--modulo the human feedback, OPAL [3]). How do the authors orient this work to those previous skill-learning frameworks? Appendix A4 compares to previous skill-based methods SpiRL and SkiMO, but these do not look like domain-generalization results. Based on Tables 1 and 2, Table 7 looks like in-domain results with train == test env. Is this correct? If so, these results do not seem particularly relevant to the paper’s argument for better domain generalization.



Experimental Results Weaknesses

(E1) Lacking architectural ablations in general. There are a lot of modules in the architecture, and it is not clear to me why each of them is necessary. Some of them, including the discrete skill selector I mentioned earlier, seem to restrict the expressivity of this skill embedding space. What is the performance of the method without predefined base skills? This seems to be the crux of the paper’s contribution, so an ablation is well-advised.

(E2) All 10 Real robot rollouts on each task have “identical starting positions.” What is the value of doing 10 rollouts with a (presumably deterministic) policy? Are object positions randomized too or made to match, as closely as possible, to the sim?



Presentation Weaknesses

(P1) Writing in several places needs work, including fixing grammar issues. Citations are not formatted properly (entirely separated by parentheses), hurting readability.

(P2) Method section in general was quite hard to understand. Notation is confusing. For instance, $x$ is not defined in equation 2. I’m assuming it is an action sequence from an expert demo. Later, in section 3.3, $\tau_t$ is suddenly introduced, and it seems to represent the same thing as $x$, except that it is a predicted action sequence. If I’m interpreting these variables correctly, perhaps a better naming would be $\tau$ for expert demo action sequence, and ${\hat{\tau}}$ for the predicted action sequence. Naming of modules needs to be made less confusing. There are 4 modules that start with “Skill,” out of 5 modules total (Figure 2).

(P3) Figure 2 references huber loss, on the reconstructed actions, but there seems to be no mention of huber loss in the paper (until the Appendix, where I see a $\| x - \hat{x} \|_2$ term). It is also unclear what the cat loss in Figure 2 refers to.

(P4) Listing equation 2 and then 4 seems a bit redundant. They look really similar besides renaming some variables. It would probably be less confusing to introduce the elbo loss in one equation instead of similar-looking equations 3 pages apart. Perhaps the loss equation 19 can replace equation 4.



References:

[1] “Parrot: Data-driven Behavioral Priors for Reinforcement Learning.” Singh et al. https://arxiv.org/pdf/2011.10024.pdf

[2] “Skill Preferences: Learning to Extract and Execute Robotic Skills from Human Feedback.” Wang et al.  https://arxiv.org/pdf/2108.05382.pdf

[3] “OPAL: Offline Primitive Discovery for Accelerating Offline Reinforcement Learning.” Ajay et al. https://arxiv.org/pdf/2010.13611.pdf

**Questions:**

(1) It seems like $w_k$ in equation 1 are learned (in training a Bi-LSTM). Where did the ground-truth action labels for Equation 1 come from? Does the learned skill embedding z-space not cluster grasping, rotation, and translation separately?

(2) It would strengthen the paper to visualize the z-space (t-SNE) and color z-space points based on which of the 3 skills they are labeled as.

(3) Is $p(z|y)$ just fitted to the action sequences in the data based on their class labels? Is it trained before the phase depicted in Figure 2? If there are two different phases as suggested by Figure 1 and 2, they should be clearly labeled as such.

(4) Is the encoder from figure 1 $q_{\phi}(z|x)$ finetuned in figure 2 as $q(z|x,y,c)$? Or are these different “encoders”?

(5) Section 3.2.2: How are two embeddings (language goal embedding and language embedding) extracted from the language instruction alone? I understand that there is a shared task embedding space in joint language + goal image embedding space, but this part of the paper is not explained well.

(6) Section 3.2.2: The authors write: “The policy $\pi(\cdot)$ should also identify the optimal base skill $y$ under the current observation.” If $y$ is already an input to the decoder, why should the decoder predict $y$ again?

(7) Looking only at the figures, how can the decoder (Skill generator) be frozen in Figure 2 if it only takes $z$ as input in Figure 1, and in Figure 2, is additionally conditioned on $y, c$?

(8) Remove the list of percentages in 4.2.1, as they are not important to the argument of the paper on generalization to new envs.

---

### Official Review · Reviewer_h9E8 · 2023-10-31

**Soundness:** 2 fair
**Presentation:** 3 good
**Contribution:** 2 fair
**Rating:** 3
**Confidence:** 4

**Summary:**

In this paper, the authors propose a novel skill-prior based imitation learning algorithm. The proposed algorithm is able to learn skill priors from unstructured data, and use those skill priors in a language conditioned imitation learning setup. The structure of the paper is the following: first, the two stage algorithm is introduced, which first learns the skill prior distribution from the play dataset, and then learns a language conditioned imitation learning policy off of the demonstrations with labels. Next, the authors present some experiments, first in a sim environment, Calvin, and then in a real robot benchmark that they created. Unfortunately, the paper ends there without much more details, such as ablation experiments.

**Strengths:**

The paper is comprehensive, showing the formulation of the skill prior informed imitation learning formulation, and learning the priors from the play data. The primary strengths of this paper are:
1. Simplifying the skill prior space. Generally, discrete skill prior based works struggle from the chicken and egg problem of classifying skill priors from data and learning them properly. By constraining the skill priors to three semantic kind of actions (translation, rotation, grasp) the algorithm makes the problem tractable.
2. Showing the algorithm scales to a real robot: a lot of time the results in simulation based papers can overfit to certain kinds of environments or quirks in the simulation, but the robot experiment show that the risk of such is not high.

**Weaknesses:**

However, there are certain major shortcomings in the evaluation in the paper and the algorithm, which are detailed below:
1. The algorithm seems very much "overfit" to the Calvin benchmark, while not being very generalizable beyond the setup. As a primary example, even the three basic "skills" seem to be overfitting to the Calvin demo behaviors, since it ignores possible robot behaviors that mix two of these skills. One easy example is opening a hinged door requires rotation and translation at the same time, which isn't covered by the algorithm's use case.
2. Another example could be the fact that the paper only focuses on skill-based manipulation algorithms, which is again a quirk of the Calvin benchmark's high-frequency control setup. However, recently there has been improvements in high-frequency controls that does not use a notion of skills, such as [1] or [2], which can be combined with learning-from-play-data algorithms such as [3] for a skill-free formulation. To show that skills are necessary for language conditioned imitation, either a comparison with such an algorithm, or a comparison on a different benchmark such as Language Table [4] would be quite useful.
3. Similarly, the real world performance is quite poor from the algorithm, which could be a case of the preset skills not really capturing the diversity of human behavior, but this question is left unanswered in the paper. The authors seemed to be content by beating out the single real baseline, HULC, which also seem to be a poor fit for the problem in hand.
4. While the language conditioning is presented as an important part of the algorithm, the "grounding" abilities are not convincing enough to show that it is a major part of the presented algorithm. Without a proper ablation experiment, this is hard to reliably conclude, which is also not presented in the paper. Similarly, ablation over the horizon may be quite important here, which is also not present in the main paper.
5. Finally, how important is extra play data if there is already sufficient language conditioned, labelled data available to learn a policy? If that is the case under which we are operating, can this algorithm still be called "learning from unstructured data"? Such questions can be answered by varying the dataset size, but because of an overdependence on Calvin as a benchmark, the authors are unable to present a real answer/experiment for this.

[1] Zhao, Tony Z., et al. "Learning fine-grained bimanual manipulation with low-cost hardware." arXiv preprint arXiv:2304.13705 (2023).
[2] Chi, Cheng, et al. "Diffusion policy: Visuomotor policy learning via action diffusion." arXiv preprint arXiv:2303.04137 (2023).
[3] Cui, Zichen Jeff, et al. "From play to policy: Conditional behavior generation from uncurated robot data." arXiv preprint arXiv:2210.10047 (2022).
[4] Lynch, Corey, et al. "Interactive language: Talking to robots in real time." IEEE Robotics and Automation Letters (2023).

**Questions:**

1. How was the horizon length of 5 decided upon?
2. As I understand, the three base skills are interchangable, so how are they labelled as "translation", "rotation", and "grasp"?
3. How large were the real world datasets?

---

> ### Author Response · Authors · 2023-11-13
>
> Thank you for the thoughtful feedback. We trust that our clarifications will enhance your understanding.
> 1. We understand your concerns regarding our algorithm potentially being too focused on the CALVIN benchmark and appearing overfitted. Contrary to this, our primary aim is to achieve a high level of generalization. During the training of our skill embeddings, the action sequences we utilized may encompass multiple base skills. Consequently, each action sequence is labeled with probabilities (e.g., 0.6 for translation, 0.2 for rotation, 0.2 for grasping). It's important to note that, in our setup, all action sequences inherently involve multiple base skills due to the continuous nature of the latent skill space. We represent base skill priors as three prior distributions for the skill embeddings. If a skill embedding closely aligns with the center of a specific base skill cluster, it signifies a higher likelihood of representing that particular base skill. Our video evidence demonstrates that our trained agent exhibits behaviors encompassing multiple base skills within a single action sequence.
> 2. We maintain that skills remain valuable in enhancing the generalization capabilities of the trained agent, given their incorporation of significant human prior knowledge. To substantiate this claim, we conducted two zero-shot experiments, encompassing both sim2sim and sim2real scenarios, as detailed in our paper. We firmly believe that the outcomes of these experiments provide compelling evidence supporting the notion that domain-invariant base skills play a crucial role in elevating the agent's generalization abilities.
> 3. We want to emphasize that our real-world experiments do not reflect poor performance, as they were conducted in a zero-shot experiment setting. The agent was never trained with real-world data, and considering the substantial differences in visual input between the simulation and real-world environments, the fact that our agent successfully completed tasks is noteworthy. However, for the purpose of assessing the generalization ability of our agent, we deliberately chose to operate in a zero-shot setting.
> 4. The effectiveness of language grounding is reflected in task completion over a long horizon. We choose base skills for their ease of language grounding in task composition, as opposed to lengthy action sequences. Performance improvements in language grounding are evident in the provided table. While horizon experiments at lengths 4 or 6 yielded suboptimal results, we selected a horizon of 5 for better performance. An update to the horizon ablation study is acknowledged in our latest version.
> 5. The significance of play data cannot be overstated. Our training dataset is composed of 99 percent play data, with only 1 percent being language-conditioned data. This substantial imbalance characterizes our approach as learning from unstructured data. Previous studies [1] [2] have addressed the question of the efficacy of introducing play data, demonstrating that it significantly enhances overall performance compared to relying solely on language-conditioned approaches. Only training with language-conditioned data poses two challenges: 1) a substantial burden in data collection, and 2) a decrease in learning efficiency. Play data, being both cost-effective and easily collectible, provides a compelling rationale for its inclusion in our approach.
>
> [1] Corey Lynch, Mohi Khansari, Ted Xiao, Vikash Kumar, Jonathan Tompson, Sergey Levine, and
> Pierre Sermanet. Learning latent plans from play. Conference on Robot Learning (CoRL), 2019.
>
> [2] Corey Lynch and Pierre Sermanet. Language conditioned imitation learning over unstructured data. Robotics: Science and Systems, 2021
>
> **Questions**
>
> 1. The horizon length of 5 is decided by our experiments that 4 and 6 do not have as good performance as 5. Also 5 is already enough to decide which base skill the action sequence belongs to. Longer action sequences might contain too complex combination of base skills.
> 2. Yes, a single action sequence can involve multiple base skills, making it stochastic rather than deterministic. For instance, (0.6, 0.2, 0.2) indicates a higher likelihood of a translation skill with some rotation and grasping. Classification is done using equation (1) in our paper, employing hardcoded procedures without a learning component. Magic weights are used to address inconsistencies in scale across different units like meters and degrees. Since these classifications may be nuanced and depend on human experience, we've chosen "magic weight" that reflects a common understanding of how these motions are typically defined.
> 3. It's crucial to highlight that we lack real-world data for training our agent. Our sim2real experiment is conducted in a zero-shot setting, signifying the absence of any real-world data in the agent's training. This circumstance elucidates the relatively suboptimal performance of the trained agent in real-world experiments.

---

### Official Review · Reviewer_WrLY · 2023-11-01

**Soundness:** 2 fair
**Presentation:** 2 fair
**Contribution:** 2 fair
**Rating:** 3
**Confidence:** 4

**Summary:**

This work proposes a skill-based language-conditioned policy. The objective is for the robot to understand human language commands, breaking down into skills to be executed consecutively. The architecture composes of a skill selector, labeler, base skill locator and generator.

**Strengths:**

1. The work considered language-conditioned skill-based policy, which is a good problem to study because language contains high level information that can be naturally broken down into skills.

2. Experiment setting: The tasks considered are unseen tasks that are not trained on during training, which is a good setting to evaluate skill learning.

**Weaknesses:**

1. Implementation of basic skills: The basic skills translation, rotation, and grasping are quite limited, as they only cover certain basic motion; they do not reflect the true distribution of real-world manipulation tasks.
- There are also quite a few existing works on using predefined skill primitives like MAPLE (https://arxiv.org/abs/2110.03655), Dalal et al. (https://proceedings.neurips.cc/paper/2021/file/b6846b0186a035fcc76b1b1d26fd42fa-Paper.pdf). How do the authors compare this work to prior works that also uses skill primitives?

2. Tasks are short horizon and limited: The tasks used in this work are very short horizon, e.g. "lift blue block". In other skill learning / skill primitive works, this could be already considered as a unit of a basic skill like lifting; there is no need to break it down into smaller units. Also, the point of using skills is to tackle those long-horizon tasks like "first lift blue block, then toggle switch". Therefore, I would consider tasks like this unable to evaluate the effectiveness of skill learning.

3. Missing baselines: the work compared with several skill-based RL works; but it fails to compare with MAPLE and Dalal et al. mentioned above.

**Questions:**

1. Implementation of basic skills: How are the three basic skills translation, rotation, and grasping implemented? Could you provide more details on how does these three skills decide their hyperparameters, e.g. how to know the translation distance or rotation angle?

2. Suppose the robot needs to learn a new skill (e.g. pouring), does the skill classifier needs to be retrained?

---

> ### Author Response · Authors · 2023-11-12
>
> Thank you for your thoughtful feedbacks. We believe the following clarifications based on the weaknesses and questions you mentioned could help you have a more clear understanding of our paper.
>
> - Within our framework, an action sequence has the flexibility to incorporate multiple base skills, such as translation and rotation, provided that the sequence is appropriately labeled with probabilities, exemplified by (0.6, 0.4, 0.0), indicating 0.6 possibilities for translation, 0.4 possibilities for rotation, 0,0 for grasping. Given the continuous nature of our skill space, it theoretically encompasses all conceivable action sequences in the real world. We posit that our definition of base skills constitutes domain-invariant knowledge, potentially enhancing overall generalization abilities in unfamiliar environments.
> - We appreciate your acknowledgment of the noteworthy papers [1] and [2]. In our assessment, the skill prior-based baseline models, SpiRL [3] and SkiMo [4], stand out as robust baselines, reflecting successful endeavors in the realm of skill-based reinforcement learning. Notably, SkiMo's author addresses limitations in [2], particularly concerning the resolution of long-horizon challenges. It is essential to highlight that these reinforcement learning approaches[1][2][3][4], lack integration with language and visual modules (directly get environment state from simulators). Consequently, their generalisation capabilities across diverse task domains and unfamiliar environments are limited. Hence, we only have a briefly comparisons with these skill-based RL approaches.
> - We posit that decomposing intricate skills, such as "lifting the blue block," into smaller units is imperative for enhancing generalization capabilities. Unlike other skill learning methodologies where skills are reliant on specific observations, our approach aims to establish domain-invariant base skills applicable across diverse environments, thereby bolstering overall generalization abilities. While many skill learning approaches leverage skills to expedite training efficiency, especially in the context of reinforcement learning (RL) with its inherent sample inefficiency, it's noteworthy that imitation learning (IL) tends to be more efficient but may succumb to overfitting, decreasing performance in unseen environments. Our paper's primary objective is to leverage these domain-invariant base skill priors not for accelerating the learning process but rather to augment the generalization ability of an IL-trained agent in diverse settings.
> - As we discussed in previous points, the RL-baselines we choose are two strong baselines in the field of skill-based reinforcement learning. Our model demonstrates impressive improvements compare to these two baselines, making us to have reasons to believe our model could also have outstanding performances compare to the papers [1] [2].
>
> **Questions**
>
> 1. All skills are represented as latent space embeddings, termed skill embeddings. Each skill embedding corresponds to an action sequence in the real world. These skill embeddings exist in a continuous latent space, making the utilized skills continuous. There is no need to learn specific distances for translation skills or precise angles for rotational skills. The agent only needs to learn the selection of skill embeddings in the latent space.
> 2. Training a new skill is unnecessary in our framework. The skills outlined in our paper are considered base skills, each with a fixed length of 5. Acquisition of novel skills, such as pouring, involves the sequential composition of multiple base skills. This learning process is guided by an imitation learning process.
>
> [1] Nasiriany, Soroush, Huihan Liu, and Yuke Zhu. "Augmenting reinforcement learning with behavior primitives for diverse manipulation tasks." *2022 International Conference on Robotics and Automation (ICRA)*. IEEE, 2022.
>
> [2] Dalal, Murtaza, Deepak Pathak, and Russ R. Salakhutdinov. "Accelerating robotic reinforcement learning via parameterized action primitives." *Advances in Neural Information Processing Systems* 34 (2021): 21847-21859.
>
> [3] Skill-based Model-based Reinforcement Learning, Lucy Xiaoyang Shi et al, CoRL 2022
>
> [4] Pertsch, K.; Lee, Y.; Lim, J. J. Accelerating Reinforcement Learning with Learned Skill Priors. Conference on Robot Learning (CoRL). 2020

---

> > ### Comment · Reviewer_WrLY · 2023-11-21
> > **Thank you for your response**
> >
> > I acknowledge the response by the authors and I appreciate the clarifications. Thank you!

---

### Official Review · Reviewer_iKk6 · 2023-11-04

**Soundness:** 2 fair
**Presentation:** 1 poor
**Contribution:** 1 poor
**Rating:** 3
**Confidence:** 3

**Summary:**

The authors present Skill Prior based Imitation Learning (SPIL), a framework for robotic imitation learning that breaks down a task into 3 base skills: translation, rotation, and grasping. The framework includes a low-level policy for generating action sequences from skills and a high-level policy that generates sequences of skills. The authors show strong performance on the CALVIN benchmark as well as a real robot using sim2real transfer.

**Strengths:**

The idea of decomposing a robotic manipulation task hierarchically using base skills is interesting and seems sound to me. Using translation, rotation, and grasping is widely applicable to many robot embodiments. The results on CALVIN are strong, demonstrating state-of-the-art performance. Any nonzero success on sim2real transfer is impressive.

**Weaknesses:**

While the basic idea of the paper seems sound, as far as I can tell, I believe it suffers from significant clarity issues. The method is complicated and has a lot of moving parts that are not fully explained. I find it difficult to evaluate the soundness and contribution of the paper due to these issues.

- In Section 3.2.1, it is not clear at all that trans, rot, and grasp correspond to groups of dimensions of the action space. The variables $x$ and $y$ are also not defined. The base skill classifier switches from $p(y = k \mid x)$ to $q(y = k \mid x)$ in the next section.
- While I generally figured out what was going on from Figure 1, I found the explanations in Section 3.2.2 fairly unclear. I also found Section 3.3 quite difficult to follow: for example, the "plan embedding" is never defined. I think the methods section could be improved by spending more time concretely explaining the authors' instantiation of skill learning rather than speaking so much in the generic terminology of variational inference.
- It is never explained how action sequences are sampled from the dataset. Wouldn't most action sequences include multiple base skills?
- If I understand correctly, the base skill locator is just an embedding lookup table for each of the 3 base skills. This could be clarified.
- Please fix the missing parentheses around references; it makes the paper more difficult to read.

**Questions:**

- How are the action sequences sampled during training? It seems to me that most action sequences would be likely to include multiple base skills, e.g., translation and grasping. How does the method deal with this?
- How were the magic scales $w_k$ chosen?

---

> ### Author Response · Authors · 2023-11-11
> **certain crucial factors to enhance the understanding of our paper**
>
> We would like to express our heartfelt gratitude for your detailed and thoughtful feedback. We believe that addressing certain crucial factors could further enhance your understanding of our paper.
>
> 1. We would like to mention that this base skill classifier $p(y=k|x)$ has nothing to do with the $q(y=k|x)$ in the next section. These are distinct concepts within our paper. The base skill classifier discussed in Section 3.2.1 serves as the method for stochastically labeling the ground truth. For instance, a distribution like (0.8, 0.1, 0.1) indicates that the given action sequence is more inclined towards a translation skill rather than rotation and grasping skills. The labelling process is hardcoded and doesn’t introduce any learning procedure.  This could also address your concern regarding the action sequences involving multiple base skills. It is important to note that our approach to labeling action sequences is not deterministic but stochastic. To enhance clarity,  renaming this section as the base skill labeling session could be more understandable. Thank you for your valuable feedback.
> 2. We posit that section 3.2.2 not only addresses the conventional terminology associated with variational inference but also serves as a platform for elucidating our innovative approach to seamlessly incorporate base skill priors into the traditional framework of Variational Autoencoders (VAE).
> 3. The embeddings outlined in our concept plan align with the definition provided in the HULC paper. While this concept is not a focal point in our imitation learning with skill priors, we acknowledge that we did not delve into it extensively. We apologize for any confusion and recognize the importance of clarifying this concept for a more comprehensive understanding.
> 4. We randomly sample action sequences from the dataset, operating under the assumption that all action sequences inherently comprise multiple base skills. This rationale underlies our stochastic labeling of the ground truth, a principle we elucidated earlier in our explanation.
> 5. In Section 3.3, we have thoroughly expounded upon the base skill locator task, utilizing bullet points to articulate its intricacies. This task is explicitly focused on discerning the positions of base skills within the latent space. Throughout the imitation learning process, the freezing of the skill embedding distribution in the latent space results in a fixed location for the three base skills. and yes, it can be seen as an embedding lookup table.
> 6. Thank you for mentioning the typo issue - missing parentheses around the references, we will fix this in our next version.
>
> **About your Questions:**
>
> 1. As elucidated in the fourth point, action sequences are randomly sampled from the dataset. The labeling of these action sequences is conducted in a stochastic manner rather than deterministically.
> 2. **Role of "Magic Weight" $w_k$**: The weight $w_k$ is introduced to address inconsistencies in scale across different units like meters and degrees. These values act as balancing factors and are determined based on our understanding of the inherent relationships between translation, rotation, and grasping. In our implementation, we have chosen the values of w_k as 1.4, 3.0, and 0.75, as detailed in the appendix.
>
>     **Why "Magic Weight":** The term "magic weight" is used to reflect the subjective nature of defining translation, rotation, and grasping. Since these classifications may be nuanced and depend on human experience, we've chosen "magic weight" that reflects a common understanding of how these motions are typically defined.